# Effect of Chlorogenic Acid on the Physicochemical and Functional Properties of Coregonus Peled Myofibrillar Protein through Hydroxyl Radical Oxidation

**DOI:** 10.3390/molecules24173205

**Published:** 2019-09-03

**Authors:** Xin Guo, Hengheng Qiu, Xiaorong Deng, Xiaoying Mao, Xiaobing Guo, Chengjian Xu, Jian Zhang

**Affiliations:** College of Food Science, Shihezi University, Shihezi 832000, China

**Keywords:** Coregonus peled, myofibrillar protein, chlorogenic acid, physicochemical structure, functional properties

## Abstract

The effects of chlorogenic acid (CA) (6, 30, and 150 μM/g protein) on the physicochemical and functional properties of Coregonus peled myofibrillar protein (MP) through oxidation using a hydroxyl radical oxidation system (0.01 mM FeCl_3_, 0.01 mM Asc, and 1 mM H_2_O_2_) were investigated. The result showed that CA inhibited the increase in protein carbonyl content but did not prevent losses in sulfhydryl and free amine contents caused by oxidation. The presence of CA also increased conformational changes in the secondary and tertiary structures of oxidized MP. Oxidized MP containing 6 μM/g CA had superior functional properties (solubility, emulsifying, foaming, and gel properties), while oxidized MP containing 150 μM/g CA aggregated, resulting in insolubility and a poor gel network.

## 1. Introduction

Protein oxidation often leads to physicochemical and functional changes during the processing of meat products, such as amino acid side-chain modification, peptide-chain cleavage, and changes in structural function [1]. These changes can affect the color, flavor, texture, and nutritional value of processed meat products, thereby reducing product quality [2]. Muscle proteins in meat products, especially myofibrillar proteins (MPs), are sensitive to reactive oxygen species (ROS). Hydroxyl radicals, as the most active free radicals, have been widely used in protein oxidation research using a hydroxyl radical oxidation system.

Polyphenols, which have an aromatic ring structure along with one or more hydroxyl groups, are among the most scrutinized natural antioxidants owing to their ability to delay or inhibit oxidative damage caused by reactive free radicals [3]. Phenolic compounds are important secondary metabolites in plants and are commonly found in fruits, vegetables, and herbs. They are divided into phenolic acids, flavonoids, and tannins, which are excellent hydrogen donors and metal chelating agents [4]. Polyphenols are used as additives to protect meat products from lipids and protein oxidation. The physicochemical structure and functional properties of proteins are reportedly affected after polyphenol treatment. Examples include the endogenous fluorescence of MP binding with gallic acid [5], emulsifying properties of α-whey protein covalently binding to EGCG [6], foaming stability of whey protein–cranberry polyphenol [7], solubility of lactoferrin binding to chlorogenic acid (CA) [8], and phenolics affecting *Rastrelliger kanagurta* surimi gel strength [9]. These results clearly showed that protein–polyphenol compounds effectively improve protein properties.

*Coregonus peled*, which belongs to the family *Salmonidae* and genus *Coregonus*, is naturally distributed in lakes and rivers in alpine regions and is a typical cold-water fish [10]. As *Coregonus peled* is high in protein and fat, among other characteristics, it is highly prone to protein oxidation during preservation, transportation, and processing, causing a decline in product quality [11]. Therefore, controlling protein oxidation is of great significance to the commercial value of *Coregonus peled***.** Chlorogenic acid, a water-soluble polyphenol, can be used as a food additive with advantageous antioxidant activity compared with other polyphenols. The polyphenols that affect changes in oxidized *Coregonus peled* MP have yet to be reported. Therefore, the main aim of this study was to explore the effects of CA on the physicochemical and functional properties of oxidized *Coregonus peled* MP.

## 2. Results and Discussion

### 2.1. Effect of Chlorogenic Acid on Oxidized MP Physicochemical Properties

Amino acids with NH or NH_2_ groups on their side chains are more sensitive to hydroxyl groups during protein oxidation, forming carbonyl derivatives by deamination. The carbonyl content of the oxidation groups was obviously higher than that of the control group (Table 1). Oxidized MP treated with different concentrations of CA in several oxidation groups showed effective inhibition of oxidation-induced increases in the carbonyl content, especially that treated with 6 μM/g CA. This was attributed to the ability of polyphenols to scavenge free radicals and chelate metal ions [4]. The oxidized MP treatment groups had different degrees of free amine loss compared with the control, with the group treated with 150 μM/g CA showing the greatest loss (Table 1). This result might be attributed to further reactions with free amine with protein carbonyl groups and the CA quinone under oxidation conditions, in which the former formed a Schiff base through a covalent effect and the latter through irreversible formation of amine–quinone adducts [5,12].

Myosin and actin contain a certain amount of sulfhydryl groups (SH), which are easily transformed into intramolecular or intermolecular disulfide bonds through attack by hydroxyl groups. As shown in Table 1, the SH contents obviously decreased in the oxidized groups compared with the control. Furthermore, adding CA to the oxidation of MP did not seem to prevent a reduction in the sulfhydryl content, while 150 μM/g CA caused the SH content to decrease further. These large SH losses might be due to thiol groups in cysteine residues being attacked by polyphenol quinones to form thiol–quinone adducts [13].

Surface hydrophobicity can reflect the extent of protein denaturation. Table 1 shows that the oxidized sample had significantly increased protein surface hydrophobicity compared with control, indicating that hydrophobic amino acid residues were exposed by oxidation, resulting in unfolded protein structure. CA can protect protein from damage to a certain degree, preventing the exposure of MP hydrophobic amino acid residues. Compared with the oxidation group, the protein surface hydrophobicity was reduced by 150 μM/g CA but still showed an increasing trend in the three CA concentration gradient groups, which might be due to a high CA concentration promoting a further oxidation-induced protein structure.

Dityrosine formation is another important indicator of protein oxidation. As shown in Table 1, oxidation increased the dityrosine content, while adding CA decreased the formation of tyrosine dimer. This phenomenon was attributed to free radicals produced in the oxidation system attacking amino acid side chain residues, such that the tyrosine monomer was oxidized to dityrosine. However, the free radical scavenging ability of the polyphenol decreased the chances of free radicals reacting with the residue [14,15]. Furthermore, 150 μM/g CA increased the formation of dityrosine, which might be attributed to the high CA concentration causing tyrosine to be more susceptible to attack by free radicals to form protein polymers. Saeed et al. postulated that hydrophobic interactions during oxidation can also lead to an increase in dityrosine [16], which is in agreement with the changes in surface hydrophobicity and dityrosine formation observed in this study.

### 2.2. Changes in Endogenous Fluorescence Spectra

Aromatic amino acids in proteins, especially highly sensitive tryptophan (Trp), can be used as endogenous fluorescent probes to investigate protein conformational changes [17]. In this study, MP showed a fluorescence emission spectrum typical of the Trp residue after excitation at 283 nm, with the maximum absorption peak obtained at 330 nm (Figure 1). The results showed that oxidation significantly increased the fluorescence intensity of MP, while adding CA decreased the fluorescence intensity. Trp residues buried in the protein core were exposed to the polar environment by oxidation, resulting in the increased fluorescence intensity. Polyphenol–protein, mainly linked by hydrogen bonds, adsorbed on the protein surface creates a protective effect that can decrease Trp residue exposure [18]. However, 150 μM/g CA formed amine–quinone adducts or thiol–quinone adducts, which might explain why this shielding effect was reduced. For MP oxidized with 6 and 30 μM/g CA, there were slight differences in the fluorescence intensity, perhaps due to exposure to the solvent in a partially or fully expanded state.

The λ_max_ value is closely related to the microenvironment in which the Trp residues are located [19], with λ_max_ > 330 nm implying that Trp residues were inside of a polar environment. In contrast, Trp residues were in a non-polar environment when λ_max_ < 330 nm. Chlorogenic acid is a water-soluble phenolic acid containing multiple hydroxyl groups that might increase the polarity of the Trp microenvironment. Furthermore, the redshift (from 330 to 332 nm) of the maximum emission caused by adding CA showed that the exposed Trp residues in the oxidized protein might interact with CA, with the Trp microenvironment turning toward the polar direction, resulting in a change in protein spatial structure.

### 2.3. FT-IR

The effect of CA on the FT-IR spectrum of oxidized MP is shown in Figure 2. Compared with the control group, the vibration absorption peaks of the other samples, amide-A (3400–3440 cm^−1^), became stronger and narrower, with a slight change in wavenumber, with the most obvious among changes observed for oxidized MP containing 150 μM/g CA. This phenomenon was possibly due changes in intermolecular forces, with CA possibly weakening intramolecular or intermolecular hydrogen bonding with oxidized MP. Furthermore, NH groups on the peptide chain and CA formed strong hydrogen bonds, resulting in an N–H stretching vibration and a decline in the absorption peak wavenumber [20].

Amide Ⅰ (1600–1700 cm^−1^) represents the C=O stretching vibration. As shown in Figure 2, there were no obvious absorption peaks for α-helixes, β-corners, and random coils in the region, which might overlap with the β-fold absorption peaks. Therefore, the region was considered to mainly consist of a β-fold secondary structure. The absorption peak intensity was increased after the oxidation and addition of CA, especially for MP oxidized with 6 μM/g CA. This might explain why MP gel prepared with 6 μM/g CA showed the maximum gel strength, because gel properties and water holding capacity can be related to an increase in β-sheet content. Furthermore, compared with the control group, the absorption peak intensity caused by amide Ⅱ (1500–1600 cm^−1^) C–N vibrations or N–H stretching was significantly increased, mainly corresponding to the higher frequency region (1530–1550 cm^−1^) of the β-fold structure. Subirade et al. [21] showed that the β-folding formed by intermolecular hydrogen bonding was the conformation necessary to maintain a stable network structure of the protein membrane. Amide Ⅲ, containing MP oxidized with 6 μM/g CA, had the strongest absorption peak (1220–1330 cm^−1^) among all samples, with the decrease in the absorption peak indicating that the phenolic hydroxyl group might be crosslinked with protein N–H groups.

### 2.4. SDS- Polyacrylamide Gel Electrophoresis (SDS-PAGE)

Electrophoresis was used to observe intramolecular and intermolecular crosslinking aggregation due to oxidation and CA addition. In the non-reducing gel (Figure 3a), some high-molecular-weight polymers appeared near the top of the separating and stacking gels that probably contained titin and nebulin. along with their proteolytic fragments or might also have been formed during MP extraction. Kroll et al. [22] previously reported the presence of various protein polymers induced by oxidized polyphenols by applying SDS-PAGE. Compared with the control group, the myosin heavy chain (MHC) was degraded significantly after oxidation. Meanwhile, polymer appeared on the upper part of separating gel and stacking gel. Consequently, it was speculated that these polymers might be mainly derived from myosin. After treating with a reducing agent (Figure 3b), MHC was recovered, while oxidized MP with a high CA content had remnant polymers. This indicated that these protein polymers were mainly formed by disulfide bonds. As for the existence of unrecovered polymers non-disulfide bonds, such as dityrosine, carbonyl–NH_2_, and other covalent linkages, which data can link to the data of sulfhydryl, free amines and dityrosine. This was similar to the report of Shuangxi et al. [23], who found that rosmarinic acid (60 and 300 μM/g) accelerated MP polymerization and that the reduction in MHC intensity was due to the formation of quinone–thiol and amine–quinone adducts.

### 2.5. Effect of Chlorogenic Acid on Functional Properties of Oxidized MP

Solubility, as the major protein functional indicator, can also reflect protein denaturation and aggregation. This study showed that oxidation reduced the protein solubility, while adding CA could delay the decrease in solubility to different extents (Table 2). This was possible because the reaction between CA and OH∙, which can scavenge hydroxyl radicals, led to inhibited protein aggregation.

Emulsification properties represent the ability of proteins to emulsify oil and water phases. As shown in Figure 4, the five group samples clearly had similar tendencies in emulsification activity (EAI) and emulsion stability (ESI). Oxidation caused the protein emulsification properties to decrease sharply. However, adding CA could improve this phenomenon, especially for MP oxidized with 6 μM/g CA, which had significantly enhanced EAI and ESI values. During the oxidation process, the intact protein structure was denatured by free radical attack, causing the degree of protein cross-linking to gradually increase and the flexibility for adsorbing fat particles on the surface to be lost [24] so that the protein oil–water interface was out of balance, causing a decrease in emulsification properties. However, the diffusion of free radicals can be prevented by thicker interfacial layers formed in protein–polyphenol conjugates owing to the addition of moderate CA in oxidized proteins [25]. The enhanced emulsifying capacity resulting from the presence of polyphenol was attributed to the interaction with the water–oil interface. Hydrogen bond formation, including the hydroxyl groups of CA and the polar headgroups of lipid, was particularly relevant to this interface interaction [26]. Furthermore, the protein emulsifying ability might be positively correlated with the solubility. The protein molecules involved in emulsification in the solution with increasing solubility improved the emulsifying ability.

The foaming properties are interfacial properties among the general functional properties of proteins. Figure 5 shows the foaming properties of oxidized protein after adding CA. The results were similar to the emulsification properties, wherein adding CA made the foaming properties of the oxidized protein better compared with the oxidized group. According to Wilde and Clark, the surface hydrophobicity, size, and structural flexibility of the surfactant influenced foam formation. Oxidation caused the protein hydrophobicity to increase significantly, resulting in a decreased solubility. With the influence of solubility occupying the leading position, the foaming properties of the protein changed dramatically. Furthermore, excessive oxidation also weakened the protein interface adsorption capacity and reduced protein–air interactions. Adding CA weakened the hydrophobic interactions of the oxidized proteins but increased the protein flexibility. Increasing the protein structural flexibility caused protein to diffuse more quickly to the air–water interface for wrapping air and, consequently, enhanced the foam formation [27]. Adding CA improved the foam properties by reducing the degree of protein oxidation and unfolding structure. Meanwhile, the emulsion stability was mainly affected by the interaction between proteins. Oxidation weakens the interaction between proteins and proteins, whereas protein–polyphenol complexes might enhance their intrinsic viscosity, producing more firm foams owing to the effective deceleration of the drainage rate [28]. Dipak et al. [29] also reported that altering the interfacial layer structure by interacting with sorption proteins was perhaps the reason for polyphenolic compounds strengthening the foam stability.

The gel strength can reflect the ability of a protein to form a gel. Table 2 shows the oxidation-decreased gel strength of MP. The gel strength of MP improved when adding 6 and 30 μM/g CA, while 150 μM/g CA was not conducive to gel formation. The formation of heat-induced protein gel was related to the cross-linking of protein intramolecular and intermolecular disulfide bonds, and other covalent bonds. Oxidation caused a decrease in gel strength, probably owing to a decrease in protein crosslinks. Adding a moderate amount of CA can enhance the crosslinking of protein molecule interactions under oxidation conditions. Balange et al. reported that nondisulfide covalent bonds formation induced by quinone and oxidized protein had a partial effect on gel strengthening [30]. However, the existence of oxidized MP with CA (150 μM/g) caused excessive aggregate formation and possibly shielded functional groups involved in the reaction, which enormously disrupted gel formation. High-content green tea extract disrupted the meat emulsion properties, which stopped disulfide bond formation because of the thiol–quinone adducts and block sulfhydryl groups, as reported by Jongberg et al. [31].

The results for cooking loss were similar to the changes in gel strength. Oxidation treatment exacerbated the gel cooking losses compared with the control group, while adding CA decreased the cooking losses. However, CA (150 μM/g) could significantly augment cooking losses for which was attributed to the weaker gel structure.

## 3. Materials and Methods

### 3.1. Materials and Chemicals

The whole material of *Coregonus peled* used in this study was purchased from Shihezi fishery in Xinjiang, China (Saihu Fishery Technology Development Co, Ltd.) and delivered to the laboratory by cold chain transport within 6 h. The white meat on the back was collected and immediately used for myofibril preparation or stored in a freezer at −80 °C until further use. Chlorogenic acid purchased from Shanghai YuanYe Biotechnology Co., Ltd. (Shanghai, China).

Homogenizer (GYB 60-6s) purchased from Shanghai Huadong High Pressure Homogenizer Factory (Shanghai, China). The high speed refrigerated centrifuge (Stratos) was purchased from Thermo Fisher Technologies (Waltham, USA). Fluorescence spectrophotometer (970CRT) was purchased from Shanghai Precision Instrument Co., Ltd (Shanghai, China). UV-visible spectrophotometer (Cary 50) purchased from Shanghai Spectrum Instrument Co., Ltd (Shanghai, China). Fourier infrared spectrometer (Nicolet iS10) was purchased from Thermo Fisher Scientific (Waltham, USA). Vortex oscillator (XH-B) purchased from Changzhou Weijia Instrument Manufacturing Co., Ltd (Changzhou, China). Texture Analyzer (TA.XT Plus) purchased from Stable Micro Systems (Surrey, UK). All of the other chemical reagents used in this study were of analytical grade and from Xinjiang Tooken Biotechnology Co., Ltd. (Shihezi, Xinjiang, China).

### 3.2. Preparation of MP

Myofibrillar protein was extracted according to methods of Lu et al. [32] with slight modifications, as described below. Fish mince was homogenized with precooled distilled water in a 1:4 (*w/v*) ratio using a homogenizer for 1 min. The mixture was then centrifuged at 8000× *g* for 10 min at 4 °C. The resulting pellet was washed with 0.3% NaCl using the blending and centrifugation conditions described above. Next, the pellet was dissolved and extracted with five volumes of chilled 20 mM Tris–HCl buffer containing 0.6 M NaCl (pH 7.0) at 4 °C for 1 h. After fully dissolving, the suspension was diluted with distilled water in a 1:4 (*v/v*) ratio and centrifuged at 10,000× *g* for 8 min at 4 °C, with the supernatant discarded. Finally, the collected pellets were redissolved in 20 mM phosphate buffer (pH 7.0) with 0.6 M NaCl and filtered through two layers of gauze to remove insoluble substances. The suspension was regarded as MP and the protein concentration was measured using the biuret method.

### 3.3. Oxidative Treatment with Chlorogenic Acid (CA)

Reaction mixtures were prepared with protein containing different CA concentrations (6, 30, and 150 μM/g protein). Mixture samples were oxidized with a hydroxyl radical oxidizing system (0.01 mM FeCl_3_, 0.01 mM ascorbic acid, and 1 mM H_2_O_2_) at 4 °C for 12 h. EDTA (final concentration, 1 mM) was added to stop the reaction. The pellet was washed twice with distilled water, which eliminated the color reaction against CA for some specific assays. The precipitate was then dissolved in 20 mM phosphate buffer containing 0.6 M NaCl (pH 7.0), and the protein concentration was adjusted for further use. Nonoxidized MP sample was used as control.

### 3.4. Measurement of Physicochemical Properties in MP

#### 3.4.1. Determination of Carbonyl Content

The degree of protein oxidation was determined from the carbonyl content, which was analyzed according to the method of Oliver et al. [33]. Briefly, the absorbance of 2,4-dinitrophenylhydrazine (DNPH)-treated protein solution was detected at 370 nm. The carbonyl content was calculated using a molecular absorption coefficient of 22,000 M^−1^cm^−1^ and measured in nmol/mg protein, calculated as follows:
$$\mathrm{carbonyl}\;\mathrm{group}=\frac{A}{\varepsilon \times b}$$
where A, *ε*, and b are the sample absorbance, molecular absorption coefficient (22,000 M^−1^cm^−1^), and cuvette light path, respectively.

#### 3.4.2. Determination of Free Amino Group (–NH_2_) Content

Free amino content of treated MP was determined using the *o*-phthalaldehyde (OPA) spectrophotometric assay described by Church et al. [34] and Lv et al. [35] with some modifications. The protein sample (200 μL) was mixed with OPA reagent (4 mL; 20% sodium dodecyl sulfate (SDS), 0.1 M borax, 0.1 M OPA, and 0.2% β-mercaptoethanol) and the absorbance at 340 nm was measured using a spectrophotometer after incubation for 2 min at 35 °C. The free amino group content was calculated from the L-leucine standard curve, with 1% SDS as blank.

#### 3.4.3. Determination of Total Sulfhydryl Content (–SH)

The total sulfhydryl content was measured using the Ellman method. Briefly, the absorbance of 5,5′-dithiobis-(2-nitrobenzoic acid) (DTNB)-treated protein solution was detected at 412 nm, with 20 mM phosphate buffer (pH 6.8) containing 0.6 M NaCl used as blank. The total sulfhydryl content was calculated using a molecular absorption coefficient of 13,600 M^−1^cm^−1^ and measured in nmol/mg protein, calculated as follows:$$(-\mathrm{SH})=\frac{A\cdot D}{B\cdot C}$$
where A, D, B, and C are the sample absorbance, dilution factor, protein concentration, and molecular absorption coefficient (13,600 M^−1^cm^−1^), respectively.

#### 3.4.4. Determination of Surface Hydrophobicity

Surface hydrophobicity was measured using the 1-anilinonaphthalene-8-sulfonic acid (ANS) method described by Li-Chan et al. [36]. A 4-mL aliquot of 0.05–0.2 mg/mL protein was mixed with 8 mM ANS (20 μL; dissolved in 10 mM phosphate buffer, pH 7.0), and the fluorescence intensity of each sample was measured after 2 min in a darkroom using excitation and emission wavelengths of 390 and 470 nm, respectively, and a slit width of 5 nm. The initial slope (S_0_) of fluorescence intensity against protein concentration was expressed as the protein surface hydrophobicity.

#### 3.4.5. Dityrosine Measurement

Dityrosine measurement was performed according to the method of Davies et al. [37]. MP solution was diluted to 1 mg/mL using 20 mM phosphate buffer (pH 7.0) containing 0.6 M NaCl, and then the specific MP concentration was detected using the biuret method. The fluorescence intensity was measured using a fluorescence spectrophotometer at 420 nm, with an excitation wavelength of 325 nm and slit width of 10 nm. The relative fluorescence value (A.U.) was expressed as the ratio of fluorescence value to protein concentration.

#### 3.4.6. Determination of Endogenous Fuorescence Spectra

Endogenous fluorescence spectra were obtained using a fluorescence spectrophotometer. MP solutions were diluted to 0.4 mg/mL using 20 mM phosphate buffer (pH 7.0) containing 0.6 M NaCl. To maximize the contribution of tryptophan residues to the emission spectra, MP solutions were excited at 283 nm at room temperature and the emission wavelength was recorded at 300–400 nm for subsequent analysis. All determinations were conducted in triplicate.

#### 3.4.7. FT–IR

Fourier transform infrared spectroscopy was used to investigate changes in the protein secondary structure. The sample solution was dialyzed at 4 °C for 24 h and freeze dried to a solid. The FT–IR spectrum of protein was recorded using the KBr pelleting method (1% *w/w* of sample in KBr) with transmittance at 4000–400 cm^−1^.

#### 3.4.8. SDS–polyacrylamide Gel Electrophoresis (SDS-PAGE)

SDS-PAGE was conducted using the method of Laemmli [38] with some modifications. Mixtures of protein samples (2 mg/mL) and an equal volume of loading buffer (0.5 M Tris–HCl, pH 6.8, 20% SDS, glycerol, 0.5% bromphenol blue, β-mercaptoethanol) were heated in a water bath at 90 °C for 5 min. A 10-μL aliquot of sample per lane was loaded onto the gel. Using 5% stacking gel and 12% separating gel, stacking gel conducted at 80 V, which was then adjusted to 160 V after entry of the the sample into separating gel. After running the glue, the gel was stained for 1 h using 0.25% Coomassie Brilliant Blue R-250 and then decolorized in solution containing 25% methanol and 7.5% acetic acid.

### 3.5. Measurement of Functional Properties of MP

#### 3.5.1. Protein Solubility

Protein solubility was measured using the method of Joo et al. [39]. Briefly, the protein solution (2 mg/mL) was centrifuged at 5000× *g* for 15 min at 4 °C, and then the protein concentration of the supernatant was determined. The solubility was expressed as the ratio of supernatant and original protein concentration, calculated as follows:$$\mathrm{Solubility}\;(\%)=\frac{the\;concentration\;of\;supernatant}{the\;concentration\;of\;original}\times 100\%$$

#### 3.5.2. Emulsification Properties

The emulsification properties were determined according to the method of Pearce and Kinsella [40]. Soybean oil and protein sample (2 mg/mL) in a 1:4 (*v/v*) ratio in a 50-mL plastic centrifuge tube were mixed at room temperature for 1 min using a vortex oscillator. A 20-μL aliquot of the emulsion was removed at 0.5 cm from the bottom of the centrifuge tube, added to 0.1% SDS solution (5 mL), and mixed well using a vortex oscillator. Sample absorbance was measured at 500 nm, denoted as A_1_, with 0.1% SDS as blank. After allowing to stand for 10 min, sample absorbance was measured at the same position by repeating the previous step, denoted as A_2_. The emulsifying activity and emulsion stability were expressed as indexes EAI (m^2^/g) and ESI (%), which was defined as
$$\mathrm{EAI}\;(m^{2}/g)=\frac{2\times 2.303}{c\times \left(1-\varphi \right)\times 10^{4}}\cdot A_{1}\cdot \mathrm{dilution}$$
$$\mathrm{ESI}\;(\%)=\frac{A_{2}}{A_{1}}\times 100\%$$
where c is the protein concentration, φ is the oil phase volume fraction (0.2), A_1_ is the 0-min absorbance at 500 nm, and A_2_ is the 10-min absorbance at 500 nm.

#### 3.5.3. Foaming Properties

The method of Poole et al. [41] with slight modification was used to determine the foaming capacity (FC) and foam stability (FS). The protein solution (20 mL, 2 mg/mL) was magnetically stirred for 30 min at 25 °C, and 10 mL of the solution was poured into a 50 mL plastic cylinder and homogenized with high-speed whipping for 1 min. The total volume of the solution and foam (V_1_) were measured immediately. After allowing to stand for 1 h at 25 °C, the change in foam volume was recorded (V_t_) and the FC and FS were calculated using the following equations:$$F_{C}=\frac{V_{1}-V_{0}}{V_{0}}\times 100\%\;\;\;\;\;\;\;\;F_{S}=\frac{V_{t}}{V_{1}}\times 100\%$$
where V_0_ is the original protein volume.

#### 3.5.4. Determination of Gel Strength and Cooking Loss

The protein solution (40 mg/mL) was placed in a 50 mL sealed glass bottle and kept about 25 mm above a heated water bath at 80 °C for 30 min. After the gel was formed, it was cooled to room temperature with running water and placed in a refrigerator at 4 °C overnight.

The prepared gel was equilibrated at room temperature for 30 min before analysis, and the partial supernatant solution was removed.

Gel strength was detected using a texture analyzer fitted with a cylindrical probe (P/0.5) at room temperature. The measurement parameters were as follows: pre-speed, 5.00 mm/s; test speed, 1.00 mm/s; post-speed, 1.00 mm/s; and trigger force, 5 g.

Cooking loss was obtained from the change in weight of the juices. This was determined by weighing a certain amount of the gel, then allowing it to sit on filter paper for 20 min to drain the juices, followed by weighing the gel again. Cooking loss was calculated using the following equation:$$\mathrm{Cooking}\;\mathrm{loss}\;(\%)=\frac{M_{0}-M_{1}}{M_{0}}\times 100\%$$
where M_0_ is the initial weight of the gel sample and M_1_ is the weight of the gel sample after juice loss.

### 3.6. Statistical Analysis

All experiments were performed in triplicate. The data were analyzed using Origin 8.5 and SPSS 17.0 software and were calculated as means ± standard deviation. Analysis of variance (ANOVA) was used to measure the significance of the main effects, and Tukey’s test was performed to determine significant differences (p < 0.05) among means.

## 4. Conclusions

The physicochemical and structural changes of oxidized MP depended on the dose of chlorogenic acid (CA). Low doses of CA can protect active groups (thiol and NH_2_) and reduce the formation of high molecular–weight polymers, such as carbonyl derivatives and dityrosine. Meanwhile, oxidized MP showed improving solubility and better emulsification, foaming, and gel properties. However, the high dose of CA formed amine–quinone and thiol–quinone adduct-modified oxidized MP, resulting in protein aggregation and a shielding effect that caused deterioration of the oxidized protein functional properties. Furthermore, MP and different CA doses, through covalent and noncovalent interactions, probably including quinone–protein, could be attributed to the change in tertiary structure. The oxidized MP secondary structure changed obviously at low doses of CA, and the oxidized MP tertiary structure changed slightly, but significantly, at low to moderate doses of CA. Further specific binding sites are yet to be studied.

## Figures and Tables

**Figure 1 molecules-24-03205-f001:**
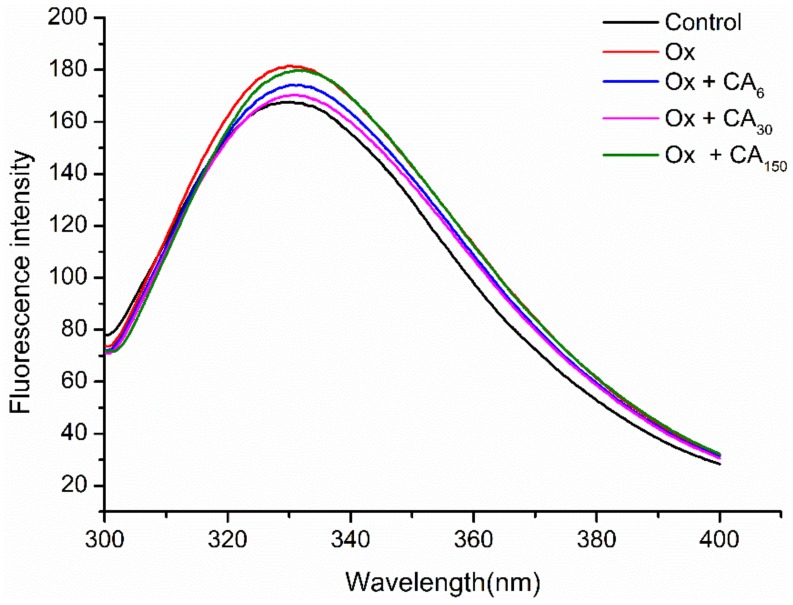
Tryptophan endogenous fluorescence of oxidized myofibrillar protein (MP) from Coregonus peled in chlorogenic acid (CA). Control: nonoxidized; Ox: oxidized; Ox + CA_6_, Ox + CA_30_, and Ox + CA_150_: oxidized in the presence of CA at 6, 30, and 150 μM/g protein, respectively.

**Figure 2 molecules-24-03205-f002:**
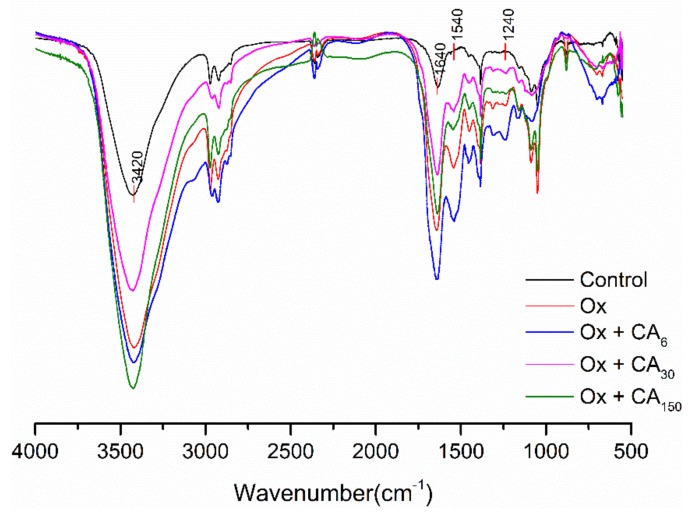
FT-IR spectra of oxidized myofibrillar protein (MP) treated with different concentrations of chlorogenic acid (CA). Control: nonoxidized; Ox: oxidized; Ox + CA_6_, Ox + CA_30_, and Ox + CA_150_: oxidized in the presence of CA at 6, 30, and 150 μM/g protein, respectively.

**Figure 3 molecules-24-03205-f003:**
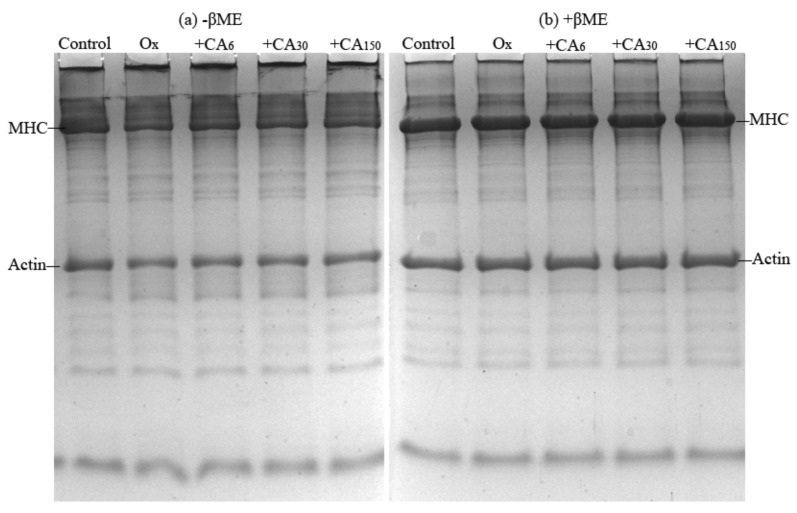
SDS-PAGE patterns of oxidized myofibrillar protein (MP) treated with chlorogenic acid (CA). Control: nonoxidized; Ox: oxidized; Ox + CA_6_, Ox + CA_30_, and Ox + CA_150_: oxidized in the presence of CA at 6, 30, and 150 μM/g protein, respectively. Samples were prepared in the presence (+βME) or absence (−βME) of 10% β-mercaptoethanol.

**Figure 4 molecules-24-03205-f004:**
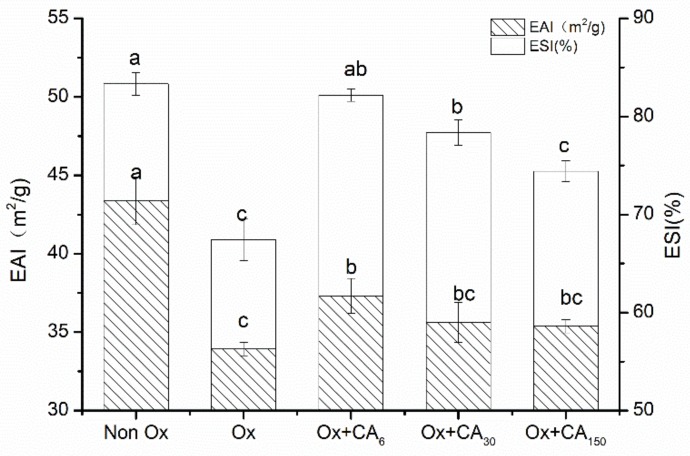
Changes in emulsifying activity (EAI) and emulsification stability (ESI) of oxidized myofibrillar protein (MP) treated with chlorogenic acid. Control: nonoxidized; Ox: oxidized; Ox + CA_6_, Ox + CA_30_, and Ox + CA_150_: oxidized in the presence of CA at 6, 30, and 150 μM/g protein, respectively. Different lowercase letters denote significant differences (p < 0.05).

**Figure 5 molecules-24-03205-f005:**
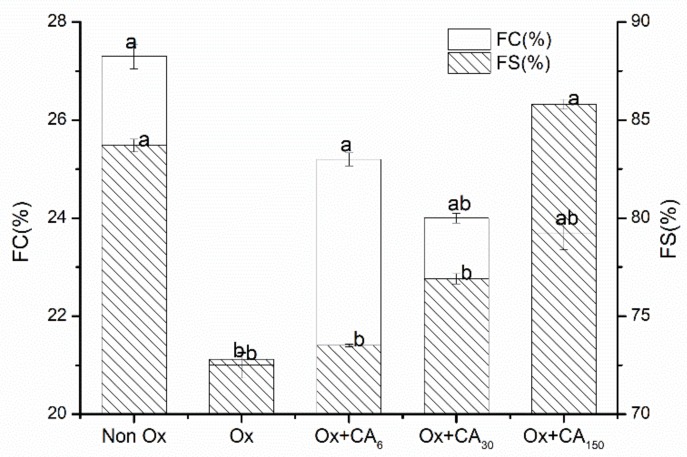
Changes in foaming capacity (FC) and foaming stability (FS) of oxidized myofibrillar protein (MP) treated with chlorogenic acid. Control: nonoxidized; Ox: oxidized; Ox + CA_6_, Ox + CA_30_, and Ox + CA_150_: oxidized in the presence of CA at 6, 30, and 150 μM/g protein, respectively. Different lowercase letters denote significant differences (p < 0.05).

**Table 1 molecules-24-03205-t001:** Changes in physicochemical and gel properties of oxidized myofibrillar protein (MP) with chlorogenic acid (CA).

Samples	Free Amino (nM/mg protein)	Carbonyl (nM/mg protein)	Thiol Groups (nM/mg protein)	Surface Hydrophobicity	Dimeric Tyrosine (A.U.)
Control	50.74 ± 0.19 ^a^	0.24 ± 0.035 ^c^	90.05 ± 1.93 ^a^	138.6 ± 2.99 ^b^	123.58 ± 6.52 ^c^
Oxidized	46.25 ± 0.16 ^d^	1.01 ± 0.008 ^a^	81.36 ± 0.65 ^b^	148.09 ± 5.06 ^a^	143.69 ± 3.62 ^a^
Ox + CA_6_	49.65 ± 0.15 ^b^	0.46 ± 0.057 ^bc^	82.39 ± 1.38 ^b^	111.66 ± 2.7 ^d^	126.92 ± 5.01 ^c^
Ox + CA_30_	48.35 ± 0.19 ^c^	0.49 ± 0.086 ^b^	81.49 ± 0.42 ^b^	125.59 ± 8.1 ^c^	134.17 ± 4.21 ^b^
Ox + CA_150_	43.43 ± 0.25 ^e^	0.52 ± 0.054 ^bc^	78.7 ± 1.05 ^c^	135.11 ± 1.77 ^b^	144.37 ± 7.8 ^a^

Control: nonoxidized; Ox: oxidized; Ox + CA_6_, Ox + CA_30_, and Ox + CA_150_: oxidized in the presence of CA at 6, 30, and 150 μM/g protein, respectively. “a–e” letters indicated significant differences in the columns (p < 0.05).

**Table 2 molecules-24-03205-t002:** Changes in solubility and gel properties of oxidized myofibrillar protein (MP) with chlorogenic acid (CA).

Samples	Solubility (%)	Gel Strength (N)	Cooking Loss (%)
Control	67.8 ± 0.4 ^a^	0.38 ± 0.12 ^ab^	41.54 ± 0.34 ^c^
Oxidized	51.8 ± 0.8 ^d^	0.34 ± 0.14 ^ab^	42.31 ± 7.04 ^b^
Ox + CA_6_	59.8 ± 2 ^b^	0.51 ± 0.20 ^a^	40.04 ± 0.84 ^d^
Ox + CA_30_	58.9 ± 0.2 ^b^	0.47 ± 0.06 ^ab^	40.38 ± 5.97 ^d^
Ox + CA_150_	57.5 ± 0.4 ^c^	0.23 ± 0.06 ^b^	50.55 ± 5.50 ^a^

Control: nonoxidized; Ox: oxidized; Ox + CA_6_, Ox + CA_30_, and Ox + CA_150_: oxidized in the presence of CA at 6, 30, and 150 μM/g protein, respectively. “a–d” letters indicated significant differences in the columns (p < 0.05).

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
