# Peer review of "Effect of Chlorogenic Acid on the Physicochemical and Functional Properties of Coregonus Peled Myofibrillar Protein through Hydroxyl Radical Oxidation"

_molecules, 2019, doi:10.3390/molecules24173205_

Round 1
Reviewer 1 Report
The work is written well. Please correct minor editorial mistakes (H2O2, units - see abstract). Please provide the source of chlorogenic acid used.
Author Response
Response to Reviewer 1 Comments
Point 1: The work is written well. Please correct minor editorial mistakes (H2O2, units - see abstract).
Response 1: Thank you very much for your good evaluation and detailed suggestion. "H2O2" has been revised "H2O2". Units "M/g" has been changed to "μM/g" in our revised manuscript which the format is Symbol. At the same time, the similar errors in the abstract have been revised in the manuscript. But the correct unit cannot be displayed on the web page state. (Page 1, Line 13-21)
Point 2: Please provide the source of chlorogenic acid used.
Response 2: Thank you very much for your valuable suggestion. We have added a supplement to this suggestion. Chlorogenic acid purchased from Shanghai YuanYe Biotechnology Co., Ltd. (Page 7, Line 262-263)
Once again, thank you very much for your comments and suggestions.

Reviewer 2 Report
Good work. In future it is worth to study regulatory and minor myofibrillar proteins as well.
Author Response
Response to Reviewer 2 Comments
Point: Good work. In future it is worth to study regulatory and minor myofibrillar proteins as well.
Response: Thank you very much for your good evaluation. As the reviewer's good advice, we will try to study regulatory and minor myofibrillar proteins in future research.
Once again, thank you very much for your comments and suggestions.

Reviewer 3 Report
The presented manuscript is well written and contains useful and interesting results regarding the effect of chlorogenic acid on the physicochemical and functional properties of oxidized Coregonus peled myofibrillar protein.
I have only a one shortcomings:
In the reference list the names of journals should be written in upper case. For example Food Chemistry and others.
Author Response
Response to Reviewer 3 Comments
Point: The presented manuscript is well written and contains useful and interesting results regarding the effect of chlorogenic acid on the physicochemical and functional properties of oxidized Coregonus peled myofibrillar protein. I have only a one shortcomings: In the reference list the names of journals should be written in upper case. For example Food Chemistry and others.
Response: Thank you very much for your good evaluation and detailed suggestion. The reference list the names of journals have been revised upper case. (Page 11-13)
Once again, thank you very much for your comments and suggestions.
